# Factors Associated with Prolonged SARS-CoV-2 Viral Positivity in an Italian Cohort of Hospitalized Patients

**DOI:** 10.3390/diseases12070138

**Published:** 2024-06-28

**Authors:** Simona De Grazia, Francesco Pollicino, Chiara Giannettino, Chiara Maria Errera, Nicola Veronese, Giovanni M. Giammanco, Federica Cacioppo, Giuseppa Luisa Sanfilippo, Mario Barbagallo

**Affiliations:** Department of Health Promotion, Mother and Child Care, Internal Medicine and Medical Specialties “G. D’Alessandro”, University of Palermo, 90127 Palermo, Italy; simona.degrazia@unipa.it (S.D.G.); francesco.pollicino@asptrapani.it (F.P.); chiara.giannettino01@you.unipa.it (C.G.); chiaramaria.errera@you.unipa.it (C.M.E.); giovanni.giammanco@unipa.it (G.M.G.); federica.cacioppo@unipa.it (F.C.); giuseppaluisa.sanfilippo@unipa.it (G.L.S.); mario.barbagallo@unipa.it (M.B.)

**Keywords:** COVID-19, prolonged positivity, viral load, autoimmune disease, chronic kidney disease

## Abstract

Clinical or microbiological factors potentially associated with prolonged COVID-19 PCR positivity are still poorly underexplored, but they could be of importance for public-health and clinical reasons. The objective of our analysis is to explore demographic, clinical, and microbiological factors potentially associated with a prolonged positivity to SARS-CoV-2 among 222 hospitalized patients. Prolonged detection positivity for SARS-CoV-2 RNA in swap samples, defined as positivity more than 21 days, was the outcome of interest. The 56 cases with a prolonged positivity to SARS-CoV-2 were matched for age and sex with 156 controls. The cases reported a significantly higher presence of diabetes mellitus, autoimmune diseases, chronic kidney diseases, and acute coronary syndrome. Moreover, the viral load was significantly higher in a period of prolonged positivity compared to a normal period. In the multivariable analysis, the presence of autoimmune diseases and chronic kidney disease were significantly associated with an increased risk of prolonged positivity as well as medium viral load or high viral load, i.e., low Ct value ≤ 30 indicating high viral load. The results of this study confirmed that in a large population of hospitalized patients with COVID-19 manifestations, the prolonged positivity of SARS-CoV-2 detection with nasopharyngeal swab was mainly related to autoimmune diseases, chronic kidney disease, and to baseline viral load.

## 1. Introduction

The COVID-19 epidemic started from Wuhan city in China toward the end of December 2019, and it was caused by a newly emerging virus [1]. Genetic studies have revealed that the novel coronavirus, known as SARS-CoV-2, belongs to the *Coronaviridae* subfamily. This family also includes other viruses, like severe acute respiratory syndrome coronavirus (SARS-CoV) and Middle East respiratory syndrome coronavirus (MERS-CoV) [2,3]. SARS-CoV-2 causes coronavirus disease 2019 (COVID-19), a respiratory disease, that WHO declared a public health emergency in January 2020 and later recognized as a global pandemic in March 2020 [4]. The COVID-19 has upended the daily life of individuals around the world, with historic numbers of cases and deaths, between patients and healthcare personnel, due to its high virulence and contagiousness [5]. COVID-19, by its nature, often leads to complications such as pneumonia, multiple organ dysfunction, renal failure, and acute respiratory distress syndrome [6].

By the end of December 2020, the COVID-19 pandemic had caused more than 67.8 million confirmed cases across 215 countries, leading to over 1.5 million fatalities. The global impact of the virus was profound, affecting public health systems and communities worldwide [7].

An infected person can generate droplets that are smaller than or equal to 10 μm. These smaller droplets have the potential to travel more than one meter; in contrast, larger droplets, produced during activities like coughing or speaking, do not typically spread beyond approximately one meter, due to gravity. In the first case the route is airborne transmission by aerosol, and research has shown that SARS-CoV-2 can remain viable in aerosols for extended periods, allowing it to spread rapidly across significant distances. Additionally, person-to-person transmission occurs through respiratory droplets containing virus particles during close contact, and via contact with contaminated surfaces. COVID-19 transmission can occur from both symptomatic patients and asymptomatic carriers [8,9]. In the older population and immunocompromised patients, SARS-CoV-2 often infects the lower respiratory tract and, in some cases, it can cause fatal pneumonia with symptoms such as cough, fever, dyspnea, myalgia, and sometimes with diarrhea [10]. The gold standard diagnostic test for SARS-CoV-2 since the start of the pandemic has been the Reverse-Transcriptase Polymerase Chain Reaction (RT-PCR), which detects viral RNA. The adoption of nasopharyngeal swabs for SARS-CoV-2 testing has undeniably contributed to more effective control of COVID-19 transmission [11]. During the COVID-19 pandemic, we observed that certain clinical conditions result not only in a poor prognosis but also lead to an extended disease course [12]. The persistence of a positive PCR test in patients does not definitively indicate the presence of live virus—maybe it could reflect residual viral RNA from inactive or dead virus particles [10].

The literature shows that there are various risk factors for long-term positive SARS-CoV-2 infection, such as age over 70, hypertension, the presence of type 2 diabetes mellitus (T2D), obesity, ischemic heart disease, chronic renal insufficiency, hematologic diseases, autoimmune diseases, immunosuppression, and chronic liver and neurological conditions [13,14,15,16,17,18,19]. Hypertension delays SARS-CoV-2 viral clearance and exacerbates airway hyperinflammation in the respiratory tract [1]. Immunosuppressed conditions include cancer, HIV, and patients with solid organ transplants (SOT) [17,20]. Patients with cancer are highly vulnerable to COVID-19 and its most harmful effects, due to the immunosuppression from their disease and treatments [21]. Age is a significant risk factor for a longer duration of positivity to COVID-19 [14]. Older individuals, especially those above 70 years of age, have been consistently shown to experience more severe illness and prolonged viral shedding compared to younger individuals [14]. Several factors contribute to this increased susceptibility and prolonged positivity among older adults: a weakened immune response: underlying health conditions, reduced lung function, and delayed immune response [14]. As people age, their immune systems tend to weaken, making it harder for their bodies to fight off infections effectively [14]. This can result in a longer duration of viral shedding and a prolonged illness course. Older adults are more likely to have underlying health conditions, such as heart disease, diabetes, and respiratory issues, which can exacerbate the COVID-19 symptoms and prolong recovery. Age-related changes in lung function, such as decreased elasticity and reduced efficiency of gas exchange, can make older individuals more vulnerable to respiratory infections like COVID-19 [14]. Prolonged viral shedding may occur due to the slower clearance of the virus from the respiratory tract [14]. Older adults may have a higher initial viral load upon infection, which can lead to a more severe illness and a longer duration of viral shedding [14]. Aging can also result in a slower immune response to viral infections, prolonging the time it takes for the body to clear the virus [14].

These patients, as well as being at increased risk of having a protracted COVID-19 infection, are also more likely to have more severe disease consequences than people without the same comorbidities [22,23,24]. The introduction of vaccines represents a significant milestone in managing SARS-CoV-2 infection. Vaccination has led to decreased disease severity and better outcomes [25]. This impact is particularly crucial for immunosuppressed patients, who often face a more challenging prognosis and limited response to conventional treatments [25].

However, the literature available so far, was made up of studies with a limited sample size.

Given this background, our analysis aims to assess the impact of several clinical and microbiological risk factors on COVID-19 PCR positivity in a large cohort of hospitalized patients in the “COVID Internal Medicine Department of University Hospital Paolo Giaccone” in Palermo, Italy from September 2020 to September 2022.

## 2. Materials and Methods

### 2.1. Study Population

All patients aged >18 years and hospitalized in the University Hospital (Policlinico) ‘P. Giaccone’ in Palermo, Sicily, Italy were enrolled from September 2020 to September 2022 [26]. The study was approved by the Local Ethical Committee during the session of the 28 April 2021 (number 04/2021).

### 2.2. Virological Investigation: Detection of SARS-CoV-2 Genome and Variants

All nasopharyngeal or oropharyngeal samples were analyzed for SARS-CoV-2 RNA using the available commercial multiplex real-time PCR assay as part of routine diagnostic. Virological investigation was conducted in accordance with the manufacturer’s instructions as follows: Seegene Allplex SARS-CoV-2 Assay, four-target assay for E, N, and S/RpRd genes (Seegene Allplex, Seoul, Republic of Korea); Roche Cobas, SARS-CoV-2 test, dual-target assay for ORF1a/b and E genes (Roche Cobas, Monza, Italy); DiaSorin Simplexa, dual-target assay for S and ORF1a/b genes (DiaSorin, Saluggia, Italy); SARS-CoV-2 ELITe MGB^®^ Kit dual-target assay for RNA-dependent RNA polymerase (RdRp) (ELITech, Torino, Italy) and ORF8 genes and Xpert^®^ Xpress SARS-CoV-2 three-target assay for E, N and RdRP genes (Cepheid, Milan, Italy). The Ct value of real-time PCR assays was used as a proxy measure of viral load (low Ct value ≤ 30 indicating high viral load).

### 2.3. Outcome: Prolonged Positivity to SARS-CoV-2

The outcome of interest was the prolonged detection of SARS-CoV-2 RNA in swap samples, defined as positivity, independently from severity, more than 21 days [12]. Participants having a positivity less than 21 days were used as controls.

### 2.4. Factors Potentially Associated with Prolonged Positivity to SARS-CoV-2

Other than demographic characteristics, such as age and sex, based on the previous literature we included as clinical factors the presence of pneumonia (defined using radiological and clinical information) and the presence of relevant chronic medical conditions [12]. In particular, the literature has suggested that diabetes mellitus, cancer (solid), autoimmune diseases, chronic kidney disease, coronary acute syndrome, and leukemias/lymphomas are the most relevant factors [14,15,16,17,18]. Even if we thought to include AIDS as potential factor, no one of the participants included suffered from this condition. HIV antibodies, when assessed in cases and controls, were negative in all the patients.

### 2.5. Statistical Analysis

Before conducting the analysis, all patient records and information were anonymized and de-identified. Continuous variables were assessed for normal distribution using the Kolmogorov–Smirnov test. Subsequently, we reported means and standard deviation (SD) values for quantitative measures, while categorical variables were expressed as percentages. This analysis was stratified based on prolonged positivity or lack thereof for SARS-CoV-2. When dealing with non-normally distributed data, we reported the median along with the interquartile range (IQR). To assess homoscedasticity of variances, we employed Levene’s test. If the assumption of homoscedasticity was violated, we utilized Welch’s ANOVA. For continuous variables, *p*-values were calculated using Student’s *t*-test, while for categorical variables, the Mantel–Haenszel Chi-square test was applied.

The relationship between the factors mentioned before and a prolonged positivity to SARS-CoV-2 was analyzed using an adjusted logistic regression and reported as odds ratios (ORs) with their 95%CI (confidence intervals). We assessed collinearity among factors using the variance inflation factor (VIF), with a threshold of two. However, none of the factors were excluded based on this criterion. All analyses were conducted using the SPSS 26.0 for Windows (SPSS Inc., Chicago, IL, USA). In our analysis, all statistical tests were two-tailed, and we considered statistical significance based on the *p*-value < 0.05.

## 3. Results

The 56 cases with a prolonged positivity to SARS-CoV-2 were matched for age (*p* = 0.58) and sex (*p* = 0.24) with 156 controls. The mean age of the patients was 63.9 years (SD = 16.9, range: 18–96), and mainly females (57.0%). The mean positivity duration to SARS-CoV-2 in cases was 33 days, versus a mean length of positivity of 10 days in the control group.

Table 1 shows the main descriptive characteristics, by presence or not of prolonged positivity to SARS-CoV-2. Among the characteristics explored, the cases did not show any significant differences in the presence of pneumonia due to COVID-19 (*p* = 0.30) or cancer (*p* = 0.28), but a significantly higher presence of diabetes mellitus (40.0% vs. 25.0%, *p* = 0.03), autoimmune diseases (12.3% vs. 4.5%, *p* = 0.04), and chronic kidney diseases (21.5% vs. 6.4%, *p* = 0.001) (Table 1). Moreover, patients with prolonged positivity showed a significantly higher prevalence of acute coronary syndrome (*p* = 0.02). Finally, patients reporting a prolonged positivity reported a higher viral load than the controls (*p* = 0.02) (Table 1).

Table 2 shows the multivariate analysis taking as a prolonged positivity to SARS-CoV-2 as outcome, defined as a positivity over 21 days. Among the factors investigated, the presence of autoimmune diseases (OR = 3.802; 95%CI: 1.218–11.870; *p* = 0.022) and chronic kidney disease (OR = 3.138; 95%CI: 1.231–7.997; *p* = 0.017) were significantly associated with an increased risk of a prolonged positivity. At the same time, medium viral load (OR = 3.280; 95%CI: 1.184–9.089; *p* = 0.022) and high viral load (OR = 3.631; 95%CI: 1.451–9.087; *p* = 0.006) are predictors of prolonged positivity to SARS-CoV-2. On the contrary, the other factors investigated were not associated with any increased risk of this outcome (Table 2). Briefly, 16 patients were affected by autoimmune diseases, mainly rheumatoid arthritis (n = 8), type 1 diabetes (n = 3), and Hashimoto’s disease (n = 5).

## 4. Discussion

Persistent positivity detection for SARS-CoV-2 RNA in swap samples has been described in patients with a broad spectrum of comorbidities. However, the clinical significance of this continuous positivity is not yet clear and thus requires further investigation. Individual detection of persistently positive PCR tests may not be diagnostically useful or practical: indeed, it is not possible to establish whether this condition is associated with a real risk of transmission, since standard lab tests are unable to differentiate between viable viruses and inactive virus residues within samples collected from a molecular swab. This could potentially cause confusion among medical professionals for the interpretation of results, the behaviors to adopt, and the precautions to take. The slow elimination of the virus is probably due to the inefficacy of the designated mechanisms which can be compromised in patients suffering from numerous comorbidities, according to the literature available thus far.

Research efforts should focus on unraveling the mechanisms of viral persistence, clarifying clinical implications, and developing guidelines for managing patients with prolonged RNA detection [27]. Collaborative studies can help address the uncertainties surrounding this phenomenon [27].

Persistent SARS-CoV-2 infections may act as viral reservoirs that could seed future outbreaks [28], give rise to highly divergent lineages [29], and contribute to cases with post-acute COVID-19 sequelae (long COVID) [30]. However, the population prevalence of persistent infections, and their viral load kinetics and evolutionary dynamics over the course of infections, remain largely unknown [31].

In this observational study involving 221 patients diagnosed with SARS-CoV-2 infection, we identified chronic kidney disease, autoimmune diseases, and viral load as the primary risk factors associated with persistent infection. Chronic SARS-CoV-2 infections not only have serious clinical implications for the individuals affected, but also pose a potential public health risk. Persistently infected hosts can serve as breeding grounds for new viral variants that may transmit more effectively and evade immunity, complicating infection control efforts [25]. Aldhaeeifi et al., showed that individuals over 70 years old, or those with hypertension, hyperlipidemia, obesity, coronary artery disease, or chronic kidney disease, are more likely to take longer to test negative for COVID-19 after an initial positive result [6]. Elderly patients and those with coronary artery disease or hypertension exhibited delayed clearance of SARS-CoV-2 RNA [14]. Going into more detail, Aldhaeefi et al., suggest that being over 70 years old decreased the likelihood of having a negative PCR result within four weeks, and Voinsky et al., demonstrated that patients over 30 years old require significantly more time to achieve their first negative PCR result compared to those under 30 years old [15]. The study underscores the influence of demographics and comorbidities on the duration of PCR positivity for COVID-19 in nasopharyngeal swabs among hospitalized patients; this insight can inform clinical practice and guide patient management during the pandemic [13,32]. Patients with coronary heart disease (CHD), decreased albumin levels, and delayed antiviral therapy experienced delays in clearing SARS-CoV-2 RNA [33]. Despite many reports of more severe COVID-19 illness in individuals with diabetes or obesity, there are limited data on whether people with T2D or obesity take longer to clear the virus, as shown by the persistent SARS-CoV-2 positivity in respiratory secretions [14]. Obesity could exacerbate the severity of the disease, due to the increased vulnerability of the population and the well-documented mechanical restrictions imposed by high body weight on the dynamics of the respiratory system [14]. The expiratory reserve volume is one of the earliest parameters affected [14]. Moreover, obesity is associated with decreased airflow, increased airway hyperresponsiveness, and a higher likelihood of developing conditions like pulmonary hypertension, pulmonary embolism, respiratory tract infections, obstructive sleep apnea, and obesity hypoventilation syndrome [14,34]. Ultimately, the physiological changes caused by obesity can lead to either hypoxic or hypercapnic respiratory failure [35]. Moriconi et al., conducted a study on the duration of viral shedding among 100 consecutive COVID-19 patients (29 of whom had obesity) admitted to Cisanello Hospital in Pisa, Italy, from 16 March to 15 April 2020 [15]. Patients with obesity showed elevated levels of circulating ferritin, C-reactive protein (CRP), and tumor necrosis factor-a (TNF-a) upon admission. They also experienced longer hospital stays and required more time, compared to non-obese subjects, to test negative for SARS-CoV-2 by PCR in nasal/oropharyngeal swabs (19 ± 8 days versus 13 ± 7 days, respectively) [15]. The pathogenetic mechanisms associated with these conditions seem to be linked, at least partially, to chronic, low-level inflammatory exposure, commonly associated with adipose tissue accumulation. This inflammatory environment often contributes to the development of metabolic and cardiovascular complications [15]. Patients with chronic kidney disease (CKD) are recognized to possess an altered immune environment, potentially leading to delayed viral clearance [35]. Among several predisposing factors, individuals with CKD may exhibit compromised immune systems and other underlying health conditions, such as hypertension and diabetes, known to elevate the risk of severe illness from COVID-19 [35].

People with diabetes who contract SARS-CoV-2 face higher risks of hospitalization, severe pneumonia, and mortality compared to those without diabetes [36,37]. Persistent high blood sugar levels can weaken both innate and humoral immunity [38,39]. Additionally, diabetes is linked to a low-grade chronic inflammatory state, which can lead to an exaggerated inflammatory response and increase the risk of developing acute respiratory distress syndrome [38,39]. Recent evidence indicates that SARS-CoV-2 can directly damage the pancreas, potentially worsening hyperglycemia and even triggering the onset of diabetes in individuals who were not previously diabetic [38,39].

A cross-sectional observational study involving 138 patients admitted with SARS-CoV-2 infection at two major regional hospitals in Scotland, UK, investigated the median time until two consecutive negative nasopharyngeal swabs for SARS-CoV-2 in an inpatient population [16].

Patients with chronic kidney disease (CKD) hospitalized with SARS-CoV-2 take longer to achieve consecutive negative results on nasopharyngeal swab reverse transcription-polymerase chain reaction (RT-PCR) tests compared to those without CKD [8]. Several studies have identified CKD as a risk factor for severe outcomes following SARS-CoV2 infection [16]. Recent studies have shown that individuals with chronic kidney disease (CKD) who contract COVID-19 face a higher risk of cardiovascular complications, including pericarditis, myocarditis, thromboembolic events, acute coronary syndrome events, and hypertensive emergencies [35]. For what concerns kidney injury, autopsy tissue samples suggest that SARS-CoV-2 can bind to, replicate in, and cause cellular damage to podocytes and proximal convoluted tubules, areas where ACE2 is highly expressed [22,38]. Recent data have described the mechanisms behind SARS-CoV-2 replication and kidney damage, highlighting the roles of innate immunity and coagulation pathways [23,39]. Indeed, SARS-CoV-2 binding to ACE2 induces depletion of CD4+ and CD8+ lymphocytes [40]. Additionally, the presence of splenic marginal zone lymphoma (SMZL) comorbidity in these patients may have exacerbated lymphocyte depletion [40]. For what concerns heart damage, in vitro models using human pluripotent stem cell-derived cardiomyocytes demonstrate that SARS-CoV-2 spreads within myocardial cells through the interaction of the activated SARS-CoV-2 spike protein and the ACE2 receptor, which is highly expressed by human cardiomyocytes [41,42]. This process involves the assembly of the virus into intracellular lysosome-like vesicles, integration of viral transcripts with cellular mRNA, disruption of cellular metabolism and genomic machinery, and ultimately leads to cardiomyocyte fusion and cell death [41,42]. Furthermore, microvascular thrombosis and subsequent myocardial cell death are the most prominent findings, primarily attributed to endothelial damage and a pro-thrombotic state [43,44].

Several reports now describe a prolonged course of COVID-19 in immunocompromised patients, primarily those with primary or acquired antibody deficiencies [45]. It is becoming clear that patients with primary or secondary humoral immunity defects exhibit suboptimal responses to natural infection or vaccination and are susceptible to persistent or chronic SARS-CoV-2 infection, highlighting the crucial role of antibodies in resolving COVID-19 [45]. Many case reports of immunocompromised patients with persistent COVID-19 have suggested that patients with hematologic malignancy and treatment with B-cell depleting therapy complicated by hypogammaglobulinemia are at particular risk for persistent viral shedding and severe COVID-19 disease [18]. Taha et al., describe two cases where persistent SARS-CoV-2 infections, specifically with the Alpha or Delta variants, were successfully cleared after treatment with the monoclonal antibody combination casirivimab and imdevimab (REGN-COV2, Ronapreve) [20]. A 55-year-old male with follicular lymphoma, who had been undergoing B-cell depleting therapy, contracted a SARS-CoV-2 infection in September 2020 that persisted for more than 200 days [18]. The second case involved a 68-year-old female with chronic lymphocytic leukemia who was on ibrutinib treatment and developed a persistent SARS-CoV-2 infection (over 290 days) [18]. Niyokuru et al., presented two cases of immunocompromised patients—one a liver transplant recipient and the other a bone marrow transplant recipient—who contracted SARS-CoV-2, showing evidence of weakened humoral immunity [46]. Both cases highlighted the crucial role of humoral immunity in resolving SARS-CoV-2 infections [46].

Most immunocompromised children with SARS-CoV-2 infection had mild disease, with prolonged viral persistence >6 weeks and moderate to high viral load [47]. Immune responses in autoimmune inflammatory rheumatic disease (AIRD) patients may be attenuated and affected by immunosuppressive treatments [48]. Choi et al., presented the case of a patient with antiphospholipid syndrome undergoing rituximab treatment who, despite completing three courses of remdesivir, intravenous immunoglobulin, and later a SARS-CoV-2 antibody cocktail, experienced a prolonged SARS-CoV-2 infection that ultimately proved fatal [49].

While prolonged viral shedding during COVID-19 has been documented in immunosuppressed patients, there are few studies specifically examining the duration of symptoms in individuals with systemic autoimmune rheumatic diseases (SARDs). An interesting study published in 2022 by Michael Dilorio et al., has studied the extended duration of COVID-19 symptoms, defined as lasting 28 days or more, in individuals with systemic autoimmune rheumatic diseases (SARDs) [50]

They analyzed data from the COVID-19 Global Rheumatology Alliance Vaccine Survey (2 April 2021–15 October 2021) to identify individuals with SARDs who reported test-confirmed COVID-19. Participants provided information on COVID-19 severity and symptom duration, along with sociodemographic and clinical characteristics. Four-hundred-and-forty-one participants with both SARDs and COVID-19 were pinpointed, averaging 48.2 years of age. Among them, 83.7% were female, with rheumatoid arthritis accounting for 39.5% of the cases. The middle point for COVID-19 symptom duration stood at 15 days, with a range of 7 to 25 days. Notably, 24.2% of participants experienced prolonged symptoms, lasting 28 days or more. Among the respondents, 9.8% reported symptoms persisting for 90 days or longer, with 42 out of 429 individuals falling into this category.

Factors linked to an increased likelihood of prolonged symptom duration encompassed were: Hospitalization for COVID-19 compared to those who were not hospitalized and experienced mild acute symptoms, comorbidity count and osteoarthritis [51,52]. Individuals with systemic autoimmune rheumatic diseases (SARDs) have experienced notable impact, potentially facing heightened susceptibility to contracting COVID-19 and encountering severe acute outcomes, including hospitalization, mechanical ventilation, and mortality [51,52].

Individuals with SARDs may be prone to experiencing prolonged COVID-19 symptoms due to various factors. These may entail an extended period of viral infection, alterations in immunity, the administration of immunosuppressive medications, or the possibility of SARD flare symptoms overlapping with those of COVID-19 [53].

Anyway, further research is warranted to explore potential correlations among immunomodulating medications, types or flare-ups of SARDs, vaccine dosages, emerging viral variants, and the persistence of COVID-19 symptoms alongside other post-acute sequelae among individuals with SARDs.

At the current state of knowledge, it can be said that there is a correlation between viral load and COVID-19. Fajnzylber et al., demonstrated that higher SARS-CoV-2 plasma viral load correlates with more severe respiratory illness, reduced absolute lymphocyte counts, elevated markers of inflammation such as C-reactive protein and IL-6, and an augmented risk of mortality [24]. Viral load significantly influenced human-to-human transmission, where a higher viral load presented an increased risk for spreading the virus to others [54]. Sokolovska et al., suggest that exceptionally high viral loads found in nasopharyngeal swabs and fecal samples could stem from the severity of COVID-19, challenges in viral clearance, and compromised immune responses attributed to advanced age, underlying health conditions, such as non-Hodgkin’s lymphoma, and the immunosuppressive treatments associated with it. This underscores the heightened risks of COVID-19 in these individuals [55].

Lee et al., in a retrospective analysis shown that prolonged RT-PCR positivity for SARS-CoV-2 is independently associated with the presence of symptoms, but not with age and co-morbidity [56].

Our work presents the main strength in the breadth and variability of the sample. In the literature, there are several case reports [57,58,59] on patients with long-term SARS-CoV-2 positivity, and few studies conducted on large samples like ours [14]. Limits of the work are: the non-discrimination between vaccinated and unvaccinated; the lack of differentiation between the variants of SARS-CoV-2; and the lack of data relating to the drug therapy of the patients. Harvey et al., show in their cohort study that patients with positive antibody test results were initially more likely to have positive NAAT results, consistent with prolonged RNA shedding, but became markedly less likely to have positive NAAT results over time, suggesting that seropositivity is associated with protection from infection [60]. The duration of protection is unknown, and protection may wane over time [60].

Indeed, conducting a study during the period between the absence of COVID-19 vaccines and their eventual introduction presents certain limitations and considerations, particularly when analyzing data from patients with comorbidities such as autoimmune diseases or chronic renal failure. During this intermediate period, the absence of vaccination among your study participants adds another layer of complexity to the analysis. It becomes challenging to attribute the prolonged positivity to COVID-19 solely to the presence of comorbidities, as the lack of vaccination could also be a contributing factor. This highlights the importance of developing a more complete understanding of the dynamics between COVID-19 infection, comorbidities and vaccination status. As vaccination status can significantly impact the severity and duration of COVID-19 infection, failing to differentiate between vaccinated and unvaccinated patients may obscure important nuances in your findings. Future studies could benefit from stratifying participants based on vaccination status to better understand how immunization influences disease outcomes, particularly in individuals with comorbidities.

Additionally, the absence of differentiation between SARS-CoV-2 variants represents another limitation. Given the emergence of new variants with potentially distinct transmission dynamics and clinical impacts, understanding their prevalence among study participants could provide valuable insights into disease progression and treatment responses.

Moreover, the lack of data relating to patient drug therapy poses challenges in interpreting the observed outcomes. Similarly, we were not able to capture information about some conditions, such as hypertension since the data about comorbidities were assessed using discharge medical records that, for administrative reasons, did not include this condition. Similarly, the study was made in a period in which vaccination against COVID-19 was poorly available and mainly in nursing homes. Finally, details about medications were overall missing.

Patients with severe comorbidities such as autoimmune diseases or chronic renal failure are often subjected to complex treatment regimens that can influence their susceptibility to infections and their subsequent response to treatment.

By capturing more detailed information on medication use, including immunosuppressive agents or therapies targeting specific disease pathways, it will be possible to better clarify the interaction between underlying conditions, pharmacological interventions and outcomes of COVID-19.

While prolonged SARS-CoV-2 detection remains a complex phenomenon, this study confirms that prolonged SARS-CoV-2 detection is more common in patients with autoimmune diseases; these autoimmune conditions may include rheumatoid arthritis, lupus, or other immune-related disorders. Furthermore, patients with autoimmune diseases often receive immunosuppressive treatments to manage their condition, and these treatments can affect the immune response, potentially leading to prolonged viral shedding. In addition, chronic kidney disease, which is prevalent in some autoimmune patients, may also contribute to prolonged positivity. For what concerns the baseline viral load, the amount of virus present at the start of infection plays a crucial role—in fact, patients with higher baseline viral loads are more likely to experience prolonged positivity. Understanding viral dynamics and individual variations is essential for predicting outcomes. In summary, understanding the association between SARS-CoV2 and autoimmune diseases, chronic kidney disease, and baseline viral load is crucial. By combining these perspectives, collaborative efforts among researchers and clinicians will provide valuable insights into managing patients effectively.

## 5. Conclusions

The results of this study confirmed that among hospitalized patients with COVID-19 manifestations of variable severity and different comorbidities, the prolonged positivity of SARS-CoV-2 detection in nasopharyngeal swab was mainly related to autoimmune diseases, probably for immunosuppressive treatments and chronic kidney disease. Moreover, the baseline viral load was another important determinant of a prolonged positivity. Further studies are needed to better understand the mechanisms that prolong SARS-CoV-2 positivity in patients with autoimmune disease, specifying whether this condition is made possible by immunosuppressive therapies or by pathogenic mechanisms related to the disease.

## Figures and Tables

**Table 1 diseases-12-00138-t001:** Descriptive characteristics by presence or not of prolonged positivity to SARS-CoV-2. The data were summarized as means and standard deviation (SD) for quantitative measures, while percentages were used for categorical variables. We calculated the *p*-values using the Student’s *t*-test for continuous variables and the Mantel–Haenszel Chi-square test for categorical variables.

Factor	Cases (n = 65)	Control (n = 156)	*p*-Value
Age	65.0 (16.1)	63.5 (17.2)	0.58
Female sex	63.1	54.5	0.24
Pneumonia	72.3	78.8	0.30
Diabetes mellitus	40.0	25.0	0.03
Cancer	13.8	9.0	0.28
Autoimmune diseases	12.3	4.5	0.04
Chronic kidney disease	21.5	6.4	0.001
Coronary acute syndrome	26.2	13.5	0.02
Leukemias/lymphomas	3.1	2.6	0.83
Medium viral load	28.6	21.4	0.02
High viral load	46.4	31.1	<0.0001

**Table 2 diseases-12-00138-t002:** Predictors of prolonged positivity to SARS-CoV-2 in hospitalized patients.

Factor	Odds Ratio	95%CI Lower	95%CI Higher	*p*-Value
Age	1.000	0.980	1.020	0.972
Female sex	1.752	0.908	3.378	0.094
Pneumonia	0.727	0.357	1.481	0.379
Diabetes mellitus	1.540	0.770	3.077	0.222
Cancer	1.375	0.493	3.836	0.543
Autoimmune diseases	3.802	1.218	11.870	0.022
Chronic kidney disease	3.138	1.231	7.997	0.017
Coronary acute syndrome	1.704	0.750	3.870	0.203
Leukemias/lymphomas	1.347	0.203	8.912	0.758
Medium viral load	3.280	1.184	9.089	0.022
High viral load	3.631	1.451	9.087	0.006

## Data Availability

Data are available upon request to the Corresponding author.

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
