# Peer review of "Factors Associated with Prolonged SARS-CoV-2 Viral Positivity in an Italian Cohort of Hospitalized Patients"

_diseases, 2024, doi:10.3390/diseases12070138_

Round 1

Reviewer 1 Report

Comments and Suggestions for Authors

            The authors analyzed some factors associated with prolonged SARS-CoV-2 infection in an Italian cohort of hospitalized patients. the mosaic of SARS-CoV-2 variants circulating in the 2020-2022 period in Lombardy. They found that the prolonged positivity of SARS-CoV-2 was mainly related to autoimmune diseases, chronic kidney disease and to baseline viral load. Some concerns should be addressed before acceptance of this manuscript. In particular, several parameters not included in this study, should be assessed, such as hypertension, HIV status, and vaccination.

Comments

1.   There was a significant association between diabetes and prolonged COVID-19, but this association was not included in Abstract as a main factor (line 28). Why?

2.   The Introduction should be focused on the study.

3.   Introduction line 26. Coronaviridae in italics.

4.   Study population. The patients enrolled in this study were from September 2020 to September 2022. An important parameter not mentioned is vaccination. What was the effect of this parameter on prolonged viral shedding?

5.   Page 3, line 135 Factors potentially associated.... The authors did not include hypertension? In addition, did all the patients know their HIV status? These two parameters are quite important for this analysis. If not available, the authors should analyze HIV-1 positivity in their samples, and any immunocompromise.

6.   There is a recent paper that may be useful to include in Introduction of Discussion. (Nature. 2024 Feb;626(8001):1094-1101. doi: 10.1038/s41586-024-07029-4.)

7.   The references need edition.

Author Response

The authors analyzed some factors associated with prolonged SARS-CoV-2 infection in an Italian cohort of hospitalized patients. the mosaic of SARS-CoV-2 variants circulating in the 2020-2022 period in Lombardy. They found that the prolonged positivity of SARS-CoV-2 was mainly related to autoimmune diseases, chronic kidney disease and to baseline viral load. Some concerns should be addressed before acceptance of this manuscript. In particular, several parameters not included in this study, should be assessed, such as hypertension, HIV status, and vaccination.

R: Thank you for the comment. We have added this as potential limitation of our findings, as follows:

“Similarly, we were not able to capture information about some conditions, such as hypertension since the data about comorbidities were assessed using discharge medical records that, for administrative reasons, did not include this condition. Similarly, the study was made in a period in which vaccination against COVID-19 was poorly available and mainly in nursing homes. Finally, details about medications were overall missing.”

At the same time no one was affected by HIV in our study.

 Comments

  1. There was a significant association between diabetes and prolonged COVID-19, but this association was not included in Abstract as a main factor (line 28). Why?

R: Thank you for the question. However, in the Abstract, we did put only factors significantly associated with a prolonged COVID-19 in the multivariate analysis.

  1. The Introduction should be focused on the study.

R: Thank you for the comment. We have tried to revise the Introduction following these indications.

  1. Introduction line 26.

R: Changed.

  1. Study population. The patients enrolled in this study were from September 2020 to September 2022. An important parameter not mentioned is vaccination. What was the effect of this parameter on prolonged viral shedding?

R: Good point. Unfortunately, the important information about vaccination was not available. At the same time, we think that this relevant limitation could have modified only partially our results since in Italian population the vaccinations against COVID-19 were available for Italian population only in the second semester of 2021. This was acknowledged in the Limitations section, as declared before.

  1. Page 3, line 135 Factors potentially associated.... The authors did not include hypertension? In addition, did all the patients know their HIV status? These two parameters are quite important for this analysis. If not available, the authors should analyze HIV-1 positivity in their samples, and any immunocompromise.

R: Good point. We have now added this point in the Methods section:

“Even if we thought to include AIDS as potential factor, no one of the participants in-cluded suffered on this condition. HIV antibodies, when assessed in cases and controls, were negative in all the patients.”

  1. There is a recent paper that may be useful to include in Introduction of Discussion. (Nature. 2024 Feb;626(8001):1094-1101. doi: 10.1038/s41586-024-07029-4.)

R: Added.

  1. The references need edition.

R: The references are now in the format of the journal, sorry for the inconvenience.

Reviewer 2 Report

Comments and Suggestions for Authors

The paper by Grazia et al entitled "Factors associated with prolonged SARS-Cov-2 viral positivity in an Italian cohort of hospitalized patients" provides interesting information but needs improvement.

L113-122.- The information in sections 2.2 and 2.3 are similar. They must be integrated into one.

Table 1. - Among the autoimmune diseases, what was the proportion of patients with rheumatoid arthritis and lupus? What other autoimmune diseases were present in the sample analyzed? Was there a difference in viral load between these groups?

L186- 359 In Table 2, indicate which medications were associated as predictors of prolonged positivity for SARS-CoV-2 in hospitalized patients.

Authors should review and comment on similar work such as:

·         Lee YH, et al. doi: 10.3855/jidc.15072.

·         Harvey RA, doi: 10.1001/jamainternmed.2021.0366.

·         Dolan SA, doi: 10.1002/pbc.29277.

·         Motoc NS, doi: 10.3390/medicina58060707.

Author Response

The paper by Grazia et al entitled "Factors associated with prolonged SARS-Cov-2 viral positivity in an Italian cohort of hospitalized patients" provides interesting information but needs improvement.

R: Thank you for your consideration. We have tried to improve our article following your indications.

L113-122.- The information in sections 2.2 and 2.3 are similar. They must be integrated into one.

R: Good point. We have merged these two sections.

Table 1. - Among the autoimmune diseases, what was the proportion of patients with rheumatoid arthritis and lupus? What other autoimmune diseases were present in the sample analyzed? Was there a difference in viral load between these groups?

R: Good point. We have fully detailed this group, as follows:

“Briefly, 16 patients were affected by autoimmune diseases, mainly rheumatoid arthritis (n=8), type 1 diabetes (n=3), Hashimoto’s disease (n=5).”

Due to the limited sample size, disease by disease, we were not able to determine the prevalence of high and low viral load in the single autoimmune conditions. Sorry for the inconvenience.

L186- 359 In Table 2, indicate which medications were associated as predictors of prolonged positivity for SARS-CoV-2 in hospitalized patients.

R: Unfortunately, this information was not available in our database. We have added this as limitation of our work.

Authors should review and comment on similar work such as:

  •  Lee YH, et al. doi: 10.3855/jidc.15072.
  •  Harvey RA, doi: 10.1001/jamainternmed.2021.0366.
  •  Dolan SA, doi: 10.1002/pbc.29277.
  •  Motoc NS, doi: 10.3390/medicina58060707.

R: All these articles are now cited in the revised form.

Reviewer 3 Report

Comments and Suggestions for Authors

In the paper “Factors Associated with prolonged SARS-CoV-2 viral positivity in an Italian cohort of hospitalized patients” the authors describe a study with 56 cases prolong COVID-19 and 156 control. They tested whether co-morbidities and viral load significantly correlated with prolong COVID-19. Overall, I thought this was a great paper and have some suggestions.

Major suggestions:

·      Methods

o   Lines 122-130 are a repeat of lines 113- 121 – Seems like the paragraph got accidentally duplicated

o   Please explain who viral loads were determined in more detail and then what makes something a normal vs high load.

o   Line 275 – “as observed in our case” – please clarify, this phrase implies this study which did not look at T cells

·      References missing

o   Lines 76-92 have no references about age-related disease

o   Line 189 – no reference listed

Comments on the Quality of English Language

Minor Suggestions:

·      Make sure you define SARS-CoV-2 and COVID-19 the first time you use them and be consistent with with capitalization. What I wrote is the correct way.

o   Line 15 – COVID-19 needs defined

o   Line 18 – define SARS-CoV-2 and fix capitalization

o   Line 344 – capitalization of COVID-19

o   Line 400 - capitalization of SARS-CoV-2

·      Minor grammar edits

o   Line 86 – Sentence starts with “age” – Capitalize the A

o   Line 121 –Xpert Xpress – says there are 3 targets but lists 4 targets. Please clarify.

o   Line 162-163 please add years – “the mean age …was 63.9 yr”

o   Line 276 – please define “these” patients. Are you referring to the CKD patients?

o   Line 306 – AIIRD should be AIRD

o   Line 315. Period after al. in et al.

o   Line 344 – replace “art al” with “et al.”

o   Lines 375 – 388 can be one paragraph

o   Lines 396-398 – Please clarify the sentence starting with “For what concern the baseline viral load”…

o   Period should be after reference and not before.

§  Line 213

§  Line 257

§  Line 261

§  Line 264

§  Line 271

§  Line 273

§  Line 277

§  Line 283

§  Line 305

§  Line 311

§  Line 317

§  Line 333-334

§  Line 338

§  Line 347

§  Line 349

·      Spacing issues

o   Line 42 – extra space before the period

o   Line 49 – add space before reference

o   Line 67 – check for extra space before the reference – this might be because justified margins though so it could be fine

o   Line 69 – add space before the reference

o   Line 100 - add space before the reference

o   Line 101-102 – combine this sentence with the previous paragraph

o   Line 294 – blue vertical line – maybe from highlighting?

·      Wording issues

o   Line 59-60 – I think the word “involves” should be replaced with “infects”

o   Line 62 – “cough” is repeated

Author Response

In the paper “Factors Associated with prolonged SARS-CoV-2 viral positivity in an Italian cohort of hospitalized patients” the authors describe a study with 56 cases prolong COVID-19 and 156 control. They tested whether co-morbidities and viral load significantly correlated with prolong COVID-19. Overall, I thought this was a great paper and have some suggestions.

Major suggestions:

  •  Methods

o   Lines 122-130 are a repeat of lines 113- 121 – Seems like the paragraph got accidentally duplicated

o   Please explain who viral loads were determined in more detail and then what makes something a normal vs high load.

o   Line 275 – “as observed in our case” – please clarify, this phrase implies this study which did not look at T cells.

o   Lines 76-92 have no references about age-related disease

o Line 189 – no reference listed.

R: Thank you so much for your careful revision. We have amended to all these points adding, in the Methods section, some details about low and high load, as requested.

Minor Suggestions:

  •  Make sure you define SARS-CoV-2 and COVID-19 the first time you use them and be consistent with with capitalization. What I wrote is the correct way.

o   Line 15 – COVID-19 needs defined

o   Line 18 – define SARS-CoV-2 and fix capitalization

o   Line 344 – capitalization of COVID-19

o   Line 400 - capitalization of SARS-CoV-2

  •  Minor grammar edits

o   Line 86 – Sentence starts with “age” – Capitalize the A

o   Line 121 –Xpert Xpress – says there are 3 targets but lists 4 targets.

o   Line 162-163 please add years – “the mean age …was 63.9 yr”

o   Line 276 – please define “these” patients. Are you referring to the CKD patients?

o   Line 306 – AIIRD should be AIRD.

o   Line 315. Period after al. in et al.

o   Line 344 – replace “art al” with “et al.”

o   Lines 375 – 388 can be one paragraph

o   Lines 396-398 – Please clarify the sentence starting with “For what concern the baseline viral load”…

o   Period should be after reference and not before.

  • Line 213
  • Line 257
  • Line 261
  • Line 264
  • Line 271
  • Line 273
  • Line 277
  • Line 283
  • Line 305
  • Line 311
  • Line 317
  • Line 333-334
  • Line 338
  • Line 347
  • Line 349
  •  Spacing issues

o   Line 42 – extra space before the period

o   Line 49 – add space before reference

o   Line 67 – check for extra space before the reference – this might be because justified margins though so it could be fine

o   Line 69 – add space before the reference

o   Line 100 - add space before the reference

o   Line 101-102 – combine this sentence with the previous paragraph

o   Line 294 – blue vertical line – maybe from highlighting?

  •  Wording issues

o   Line 59-60 – I think the word “involves” should be replaced with “infects”

o   Line 62 – “cough” is repeated

R: Thank you so much for your careful revision. We have amended to all these points that further improved the quality of our work

Round 2

Reviewer 1 Report

Comments and Suggestions for Authors

    The authors addressed satisfactorely the concerns of the reviewers.

Author Response

Thank you so much!

Reviewer 2 Report

Comments and Suggestions for Authors

L23-24.- In the abstract, show in parentheses what a moderate or high viral load means.

L88.-Correct all typos.

L426-430. The authors are requested to revise their statement in the conclusion, which states: The results of this study confirmed that in a large population of hospitalized patients with COVID-19 manifestations of variable severity and different comorbidities, prolonged positivity of "Detection of SARS-CoV-2 in nasopharyngeal swabs" was mainly related to autoimmune diseases, probably for immunosuppressive treatments and chronic kidney disease. However, the authors did not perform a calculation to determine the size of the hospitalized population during the study period with prolonged positivity. Instead, they selected patients with prolonged positivity to represent a "large population (?)"

Author Response

L23-24.- In the abstract, show in parentheses what a moderate or high viral load means.

R: Added.

L88.-Correct all typos.

R: Done.

L426-430. The authors are requested to revise their statement in the conclusion, which states: The results of this study confirmed that in a large population of hospitalized patients with COVID-19 manifestations of variable severity and different comorbidities, prolonged positivity of "Detection of SARS-CoV-2 in nasopharyngeal swabs" was mainly related to autoimmune diseases, probably for immunosuppressive treatments and chronic kidney disease. However, the authors did not perform a calculation to determine the size of the hospitalized population during the study period with prolonged positivity. Instead, they selected patients with prolonged positivity to represent a "large population (?)"

R: We fully agree with this request. We have now modified the conclusions, as follows: 

The results of this study confirmed that among hospitalized patients with COVID-19 manifestations of variable severity and different comorbidities, the prolong positivity of SARS-CoV-2 detection in nasopharyngeal swab was mainly related to autoimmune diseases, probably for immunosuppressive treatments and chronic kidney disease. Moreover, the baseline viral load was another important determinant of a prolonged positivity. Further studies are needed to better understand the mechanisms that prolong SARS-CoV-2 positivity in patients with autoimmune disease, specifying whether this condition is made possible by immunosuppressive therapies or by pathogenic mechanisms related to the disease.